# Computed Tomography-Based Navigation System in Current Spine Surgery: A Narrative Review

**DOI:** 10.3390/medicina58020241

**Published:** 2022-02-05

**Authors:** Nao Otomo, Haruki Funao, Kento Yamanouchi, Norihiro Isogai, Ken Ishii

**Affiliations:** 1Department of Orthopaedic Surgery, School of Medicine, International University of Health and Welfare, Chiba 286-8520, Japan; naoootomo@yahoo.co.jp (N.O.); yamaken0331@gmail.com (K.Y.); n.isogai0813@gmail.com (N.I.); 2Department of Orthopaedic Surgery, International University of Health and Welfare Mita Hospital, Tokyo 108-8329, Japan; 3Department of Orthopaedic Surgery, International University of Health and Welfare Narita Hospital, Chiba 286-8520, Japan

**Keywords:** computed tomography-based navigation, pedicle screw, percutaneous pedicle screw (PPS), minimally invasive spinal treatment (MIST), minimally invasive spine surgery (MISS), minimally invasive spinal stabilization (MISt)

## Abstract

The number of spine surgeries using instrumentation has been increasing with recent advances in surgical techniques and spinal implants. Navigation systems have been attracting attention since the 1990s in order to perform spine surgeries safely and effectively, and they enable us to perform complex spine surgeries that have been difficult to perform in the past. Navigation systems are also contributing to the improvement of minimally invasive spine stabilization (MISt) surgery, which is becoming popular due to aging populations. Conventional navigation systems were based on reconstructions obtained by preoperative computed tomography (CT) images and did not always accurately reproduce the intraoperative patient positioning, which could lead to problems involving inaccurate positional information and time loss associated with registration. Since 2006, an intraoperative CT-based navigation system has been introduced as a solution to these problems, and it is now becoming the mainstay of navigated spine surgery. Here, we highlighted the use of intraoperative CT-based navigation systems in current spine surgery, as well as future issues and prospects.

## 1. Introduction

Spinal instrumentation surgery has continued to make extraordinary progress with the development of surgical techniques and spinal implants. In particular, spinal implants such as pedicle screws (PS) have developed remarkably, especially in the last two decades. In addition, minimally invasive spine stabilization (MISt) surgery is becoming a popular procedure against the backdrop of an aging society [1]. However, even if high-performance implants are developed, they cannot be utilized effectively if they are not placed correctly. The deviation of a spinal implant not only loses its mechanical effectiveness, but also damages nerves, blood vessels, and organs, resulting in serious complications. In recent years, computer-assisted navigation (CAN) has been introduced to ensure accurate and effective implant placement and to improve safety. CAN provides surgeons with more confidence by providing a three-dimensional (3D) visualization of the skeletal anatomy not clearly evident through surgical exposure alone, especially in complex spine surgery [2]. It is also useful for making the implant placement in the MISt procedure safer and easier without exposing anatomical landmarks [3]. In this review article, we report on the current status and application of CAN in the field of spine surgery based on previous literature, in addition to discussing future challenges and prospects.

## 2. Method

Based on previous literatures, we summarize the current status of CAN in the field of spine surgery.

## 3. Discussion

### 3.1. Trends in Spinal Navigation Systems

Navigation systems were first reported in the field of neurosurgery during the 1970s and 1980s. In 1979, the Brown-Roberts-Wells stereotactic system combined computed tomography (CT) images with a stereotaxic frame in neurosurgery to enable a highly accurate guidance for locating lesions. In 1986, Roberts et al. [4] reported the use of preoperative CT images for microsurgery of brain tumors. In the 1990s, navigation systems began to be introduced in spine surgery. In the 2000s, intraoperative CT images were reconstructed in 3D visualization and used for navigation systems. There are several advantages of intraoperative CT-based navigation, including its high reproducibility due to intraoperative CT images obtained in the actual surgical position, usefulness of highly accurate 3D reconstructed images for precise implant placement, and the ability to confirm implant placement without moving to the CT room [5].

In the O-arm^®^ Surgical Imaging System (Medtronic, MN, USA) (Figure 1), the X-ray tube and flat-panel detector (FPD) rotate 360° inside the gantry, and 3D scanning can be performed in approximately 13 s. The system can acquire highly accurate intraoperative images and create 3D reconstructed images in a short amount of time. Some advantages of the O-arm^®^ system include its ability to provide more accurate image data compared to preoperative CT navigation and capability to automate registration to rid of lengthy registration work. The imaging direction is changed by moving the tube and FPD in the gantry frame, eliminating the need for cumbersome movements during operations. In addition, since the gantry can be moved and the imaging direction can be changed at the touch of a button, it is possible to operate the system while maintaining a clean environment in the operating field [6]. The highly accurate 3D reconstructed images allow for more accurate placement of implants, which significantly improves the safety of spine surgery (Figure 2).

### 3.2. Accuracy, Complication Rate, Cost-Effectiveness, and Radiation Exposure in Navigated Spine Surgery

There are many reports on the accuracy of PS placement under the use of intraoperative CT-based navigation systems [7,8]. Van de Kelft et al. [9] reported a deviation rate of 1.8% for thoracic, lumbar, and sacral PSs inserted under intraoperative CT-based navigation. Scheufler et al. [10] reported that the accuracy rate of PS insertion in cervical and upper thoracic spine surgery under intraoperative CT-based navigation was 99.3% for the cervical spine, and 97.8% for the thoracic spine. According to a systematic review by Shin et al. [3], there were significantly fewer PS deviations under intraoperative CT-based navigation compared to the freehand technique, and neurological complications were not observed in any of the 4814 PSs inserted under intraoperative CT-based navigation, whereas they occurred in three of 3725 PSs inserted under freehand guidance. Shin et al. [3] also reported that 94% of pedicles screws were inserted accurately with navigational techniques, while 85% were inserted accurately with freehand techniques. Verma et al. [11] also reported 93.3% of the pedicle screws were inserted accurately with navigational techniques, while 84.7% were inserted accurately with freehand techniques. Yson et al. [12] compared intraoperative CT-based navigation with the freehand technique and reported significantly fewer intervertebral joint injuries (4% vs. 26.5%).

In a meta-analysis comparing intraoperative CT-based navigation and fluoroscopy, intraoperative CT-based navigation demonstrated a significantly shorter operation time and significantly lower rates of PS deviation and perioperative complications [12]. In addition, in a report comparing freehand PS insertion, there was significantly less PS repositioning due to PS deviation in intraoperative CT-based navigation [13]. Moreover, spine surgery using intraoperative CT-based navigation has been shown to reduce blood loss and complications [14,15]. Several studies have reported a significantly lower reoperation rate in spine trauma surgery using intraoperative CT-based navigation [3,13,16]. In one study, 0% of spine surgeries required reoperation in the intraoperative CT-based navigation group compared to 4.4% in the non-navigation group [17]. Furthermore, reinsertion was only required in 0.99% of 1148 cases in cervical, thoracic, and lumbar spine surgeries using intraoperative CT-based navigation [18], suggesting the high accuracy and safety of spine instrumentation using intraoperative CT-based navigation.

Accurate implant placement may be cost-effective as well, as it decreases the risk of perioperative complications and reoperation. Watkins et al. [19] reported cost savings of USD 71,286 per 100 patients in spine surgery using intraoperative CT-based navigation compared to cases without using navigation. They also reported that performing more than 254 surgeries per year for adult spinal deformities was significantly more economical in intraoperative CT-based navigation than the freehand technique [20]. This may contribute to the low reoperation rate of navigated surgery. Although there will be an initial cost in introducing the navigation system, it may be cost-effective in the long run due to the reduced risk of perioperative complications and the lower reoperation rate.

Special attention should be paid to intraoperative radiation exposure in spine surgery. Spine surgeons are frequently exposed to radiation in their daily practice. It has been reported that even low doses of radiation exposure can cause late onset radiation cataracts [21]. Long-term cumulative radiation exposure has also been reported to be a potential cause of malignant tumor [22]. Spine surgery under a navigation system has been reported to reduce the radiation exposure of the surgical team compared to conventional X-ray fluoroscopy [19].

In addition to the surgical team, intraoperative radiation exposure should also be minimized for patients. Mendelsohn et al. [23] reported that the radiation exposure to patients in spine surgery under intraoperative CT-based navigation was about 2.7 times higher than fluoroscopy. On the contrary, when fluoroscopy was frequently used in long-range spinal fusion, the radiation exposure was not significantly different from that of a CT-based navigation system [24,25,26]. The incident surface dose, which is a measure of intraoperative radiation exposure to the patient during percutaneous pedicle screw (PPS) insertion, has been reported to be 365 mGy under CT and 571 mGy under conventional fluoroscopy [27]. Some reports state that there is no significant difference in the exposure dose between intraoperative CT-based navigation and conventional fluoroscopy, while others describe a significantly lower exposure dose in navigated surgery [23]. The upper limit of patient exposure for treatment is less than 2000 mGy, and the exposure dose from intraoperative CT scanning is within the acceptable range. However, minimizing radiation exposure should always be attempted, and multiple intraoperative imaging should be avoided.

### 3.3. Application of Navigation Systems in Cervical Spine Surgery

For insertion of PS and lateral mass screws in the cervical spine, previous reports have shown that the probability of deviation ranges from 2.5% to 29.1% [28,29,30,31], and the rate of deviation is higher than that of other spinal levels. One of the reasons for this is the narrower screw insertion point and pathway of the cervical spine compared to the thoracolumbar spine. Malposition of cervical screws may not only lead to spinal cord and nerve root injuries but also vertebral artery injuries; thus, screw deviation can cause serious complications. The use of an intraoperative CT-based navigation system allows for more accurate screw placement and ensures safety (Figure 3). Kotani et al. [30] reported that the screw deviation rate was significantly lower under CT navigation guidance than the freehand technique. Ishikawa et al. [32] also reported that the intraoperative CT-based navigation system enabled more accurate PS insertion in the cervical spine than the freehand technique. However, they also reported that 2.8% of cervical pedicle screws deviated between 2 mm and 4 mm even when the intraoperative CT-based navigation system was used. The cause of cervical screw deviation is thought to be the high flexibility of the cervical spine. Although intraoperative CT images can be acquired intraoperatively, they are not actual real-time images. Therefore, it is not possible to respond to changes in alignment that may occur during the procedure. Even a slight intraoperative load can easily change the cervical alignment, and deviation is likely to occur in the cervical spine where the insertion point of the screw is narrow. It is important to feel potential changes in the cervical alignment during screw insertion, in the same way as the freehand technique. The accuracy of the position should be confirmed by comparing the actual position with navigation images by occasionally touching the surface of the lamina or spinous process with the pointer of the navigation system. In addition, in cervical spine surgery, it may be useful to obtain another intraoperative CT image after screw insertion to confirm the position of the screw.

### 3.4. Application of Navigation Systems in Scoliosis Surgery

In the case of PS insertion in scoliosis surgery, previous literature had reported deviation rates ranging from 1.7% to 15.7% [33,34,35,36,37]. Freehand insertion of the PS in scoliosis surgery is likely to deviate because the anatomical pathway is often different from normal anatomy. There is a particularly high rate of deviation in the concave T4-9 region, which is associated with a high risk of injury to the surrounding great vessels [38]. In this regard, navigated screw insertion is useful to improve safety in scoliosis surgery. Ughwanogho et al. [14] compared CT-based navigation and freehand techniques for thoracic PS insertion in idiopathic scoliosis and found a deviation rate of 0.6% for the former and 4.9% for the latter. Other literature has also shown high accuracy in CT-based navigation for PS insertion in scoliosis surgery, with 98.9% accuracy in idiopathic scoliosis and 99.3% accuracy in congenital scoliosis [39,40]. The CT-based navigation system is considered one of the most important imaging assistive technologies, especially in scoliosis surgery with congenital malformations or severe deformities (Figure 4a,b). On the other hand, radiation exposure remains an issue in young subjects; therefore, indications should be carefully considered.

### 3.5. Application of Navigation System in MISt Procedures

Navigation is also useful in minimally invasive spine surgery (MISS), such as MISt. Navigation allows the surgeon to realize the correct trajectory of spinal implants even in a limited field of view, which enables an accurate placement of the implant. Unlike the conventional fluoroscopic technique, no guidewire is required in PPS placement using the navigation system. Under navigation, the direction is confirmed with a probe, the position of the skin incision is determined, and the probing and tapping operations are performed. At this time, the trajectory can be left on the navigation monitor so that the direction of screw insertion is not lost even without a guidewire. In addition, PPS can be inserted into the upper thoracic vertebrae, which is difficult to visualize under fluoroscopy.

In lateral lumbar interbody fusion (LLIF), frequent fluoroscopic confirmation is required during intervertebral manipulation. In contrast, the use of the navigation system almost eliminates the need for intraoperative fluoroscopy. Especially in cases of severe degeneration or where anatomical positioning is difficult to confirm, the direction of the intervertebral space can be determined three-dimensionally. In recent years, it has been reported that LLIF and PPS placement were performed entirely in the lateral recumbent position [41], which resulted in shorter operative times and reduced costs [7,42]. However, when LLIF and PPS are performed in the right lateral decubitus position, the devices may interfere with the bed and trunk-holding devices, especially when inserting the PPS on the right side. Therefore, it is necessary to consider the patient’s position and the position on the bed before the procedure.

The S2 alar iliac (S2AI) screw placement also has advantages when using the navigation system (Figure 5a,b). S2AI screw fixation is used as a strong distal anchor in various spinal diseases such as spinal deformities, spinal tumors, and infections [43,44]. On the other hand, it is not easy to insert a S2AI screw of sufficient length in a precise direction. When inserting a S2AI screw under fluoroscopy, precise control is required for obtaining fluoroscopic views such as the frontal view and teardrop view. Moreover, frequent use of fluoroscopy increases radiation exposure and can potentially cause contamination problems. The deviation rate for freehand S2AI screw insertion has been reported to be 6.2% [41]. Using the navigation system, it is possible to place a screw of sufficient length while confirming the correct insertion direction. The navigation system is also useful when the anatomical positioning is unclear under fluoroscopy. Nottmeier et al. [45] reported that 20 patients who underwent navigated S2AI insertion had no screw deviation and complications. In other reports, the accuracy rate of the S2AI screw insertion under CT-based navigation generally exceeded 95% [46,47].

In addition to spinal instrumentation, the advantages of CT-based navigation can be applied to surgery for compressive and neoplastic lesions. In our case, we could safely and accurately perform decompression or resection of ossification of posterior longitudinal ligament or spinal tumor by utilizing navigation system. The use of navigation technology has the potential to make complex spine surgery less invasive and more accurate.

### 3.6. Robotics-Assisted Surgery

Preoperative planning software and robotic devices improve the feasibility, accuracy, and efficiency of surgery by facilitating the placement of PSs, especially in anatomically difficult patients. The purpose of the robot functions as a semi-active surgical assistive device is not to take the place of, but to provide a variety of tools that can expand, the surgeon’s ability to treat patients. Although robotic systems from various companies exist, the principles of robotically guided PS insertion are the same regardless of the system used. There are some robotic systems including Mazor X Stealth^TM^ Edition Robotic Guidance System (Medtronic, MN, USA) for spine surgery, the ROSA^®^ (Medtech, Montpellier, France), the ExcelsiusGPS^®^ robot (Globus Medical, PA, USA), and the SurgiBot and ALF-X^®^ Surgical Robotic systems (both from TransEnterix, NS, USA). PS installation under robotic guidance has the following advantages over the traditional dorsal instrumentation technique: increased accuracy and safety of pedicle screw insertion [48,49,50]; accuracy in screw size selection and planned screw placement [48]; reduced radiation exposure to surgeons, patients, and operating room staffs [51,52,53,54,55,56,57]; simplicity, ease of use, and shallow learning curve [58,59,60]; easy registration and reduced operative time [48]; significantly improved surgeon ergonomics and dexterity in the repetitive task of PS placement [61,62,63]; expanded range of functions to enable use in minimally invasive surgery [52,64]. Robotic-guided PS insertion has been reported to be 94.5–99% accurate in studies of complex deformities and re-operations, including congenital malformations, degenerative diseases, destructive tumors, and trauma [48,49,50,51,52,53,58,65]. The safety of this technique, in terms of reduction of complications and intraoperative radiation exposure, is an important factor in the success of the procedure. The feasibility of this technique has been extended to minimally invasive procedures and cervical use while replicating its benefits. The technique has been used with consistent success in 25–30 patients and has a reasonable learning curve [66]. Initially used primarily for thoracolumbar PS insertion, the latest robotics and software have been adapted for use in the cervical spine with equal efficiency and accuracy. The challenges include radiation exposure, screw malposition, equipment and software failure, registration failure, time, high cost, and learning curve [66].

### 3.7. Limitations of Navigation Systems

Limitations of the navigation system include the possibility of deviation of the implant without noticing the misalignment of the registration and the flexibility of the spine, as well as the possibility of contamination of the operative and clean fields during the navigated surgery. Intraoperative CT images are not real-time images, and the implant position on the navigation monitor may differ from the actual patient position, especially in surgeries where the position and alignment of the vertebrae tend to move intraoperatively, such as cervical spine surgery [67]. Another unique problem is that the accuracy of the navigation system may decrease as the distance from the reference frame increases, and caution must be taken when performing long-range fixation procedures. In addition, in corrective spine surgeries, real-time visualization on the navigation monitor is impossible during corrective procedures; therefore, the usage of fluoroscopy should also be considered.

In addition, obtaining intraoperative CT images takes time, and the space in the surgical field is limited. Surgeons and surgical teams should become accustomed to operating the navigation systems. Rivkin et al. [68] compared 30 cases of the first half and 30 cases of the second half of CT navigation surgery and reported that PS deviation was significantly reduced in the latter. The surgeon should also consider the limitations and problems of the navigation systems and use them as one of the supporting tools in spine surgery.

### 3.8. Future Perspective and Challenges

It is now possible to perform spinal instrumentation surgery accurately and safely by using navigation systems. However, even in current systems, the margin of error for positioning has not been reduced to less than 1 mm. Another issue that needs to be improved in the future is the fact that the intraoperative CT images are not in real time; thus, they cannot respond to intraoperative alignment changes. Furthermore, errors are also likely to occur when there is a certain amount of distance between the reference frame and surgical site. There are no alerts when errors or misalignments occur, which may lead to serious accidents if the surgeon is not aware of these pitfalls. An alert system in the navigation device may be able to mitigate the problem and provide increased safety. Augmented reality (AR) may be one effective system as a solution to these problems. The results showing its efficacy and safety have been reported in actual clinical practice [69,70].

The cost of installing the navigation device and peripheral equipment is also a major issue for the widespread use of navigated surgery. In addition, although the mobility of the device is improving, the number of operating rooms where it can be used is limited due to its size and weight. The development of more compact and lightweight navigation devices is also desired.

Another prospect for intraoperative navigation is the use of AR and robotics-assisted surgery. If navigation images are displayed on a head-up display (HUD), the surgeon does not need to take his eyes off the surgical field to look at the monitor. We believe that this technology can also be applied to training of navigated surgery for young surgeons.

## 4. Conclusions

We described the current status and application of navigation systems in the field of spine surgery based on previous literature. Navigation is useful for accurate and safe implant placement, and it is also expected to be cost-effective by lowering the reoperation rate. However, overconfidence in navigation should be avoided, and it should be utilized as one of the supporting imaging technologies. It is necessary for surgeons to have sufficient knowledge of anatomy and basic surgical techniques, and to recognize the existence of specific pitfalls in the navigated surgery. We look forward to the further development and improvement of navigation systems in the future.

## Figures and Tables

**Figure 1 medicina-58-00241-f001:**
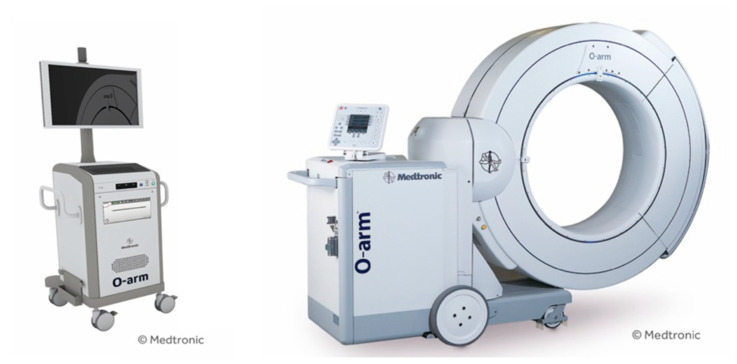
O-arm^®^ and navigation device. The X-ray tube and flat-panel detector (FPD) rotate 360° inside the gantry, and three-dimensional (3D) scanning can be performed in approximately 13 s. High-precision 3D reconstruction is possible in a short amount of time.

**Figure 2 medicina-58-00241-f002:**
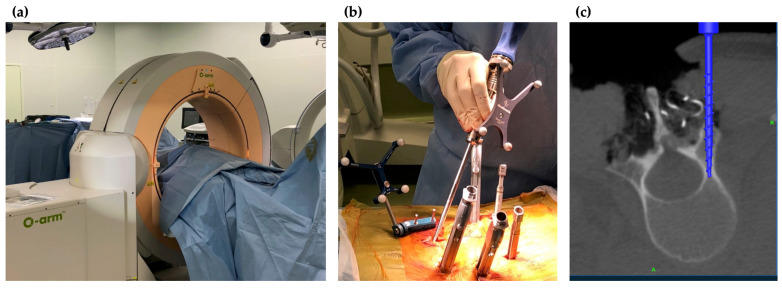
Intraoperative images of O-arm^®^ navigation. (**a**) Setting of O-arm^®^; (**b**) insertion of pedicle screws under CT navigation; (**c**) reconstruction CT image on the navigation monitor.

**Figure 3 medicina-58-00241-f003:**
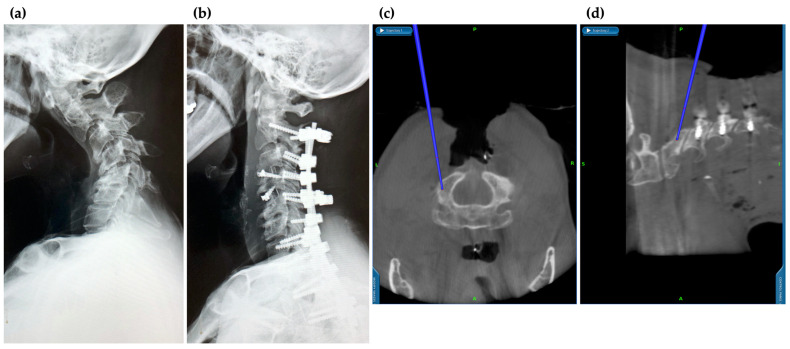
Application of navigation in cervical spine surgery. The surgical procedure can be performed safely while checking the vertebral artery running in the vicinity. Because of the high flexibility of the cervical spine, intraoperative alignment changes must be carefully monitored. (**a**) Pre-operative X-ray; (**b**) post-operative X-ray; (**c**) intraoperative axial; and (**d**) intraoperative sagittal views on navigation monitor.

**Figure 4 medicina-58-00241-f004:**
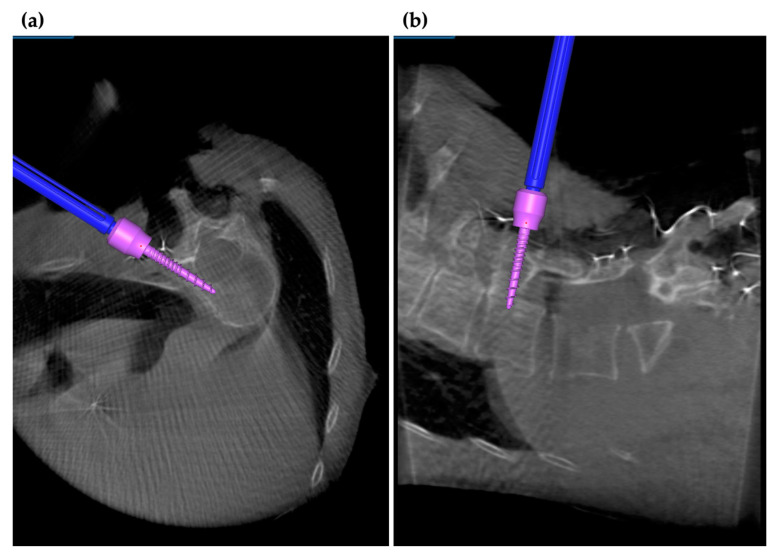
Navigation in scoliosis surgery. In scoliosis surgery, where the direction of the pedicle is difficult to decipher, intraoperative navigation can be used to reduce PS displacement. (**a**) Intraoperative axial and (**b**) intraoperative sagittal views on navigation monitor.

**Figure 5 medicina-58-00241-f005:**
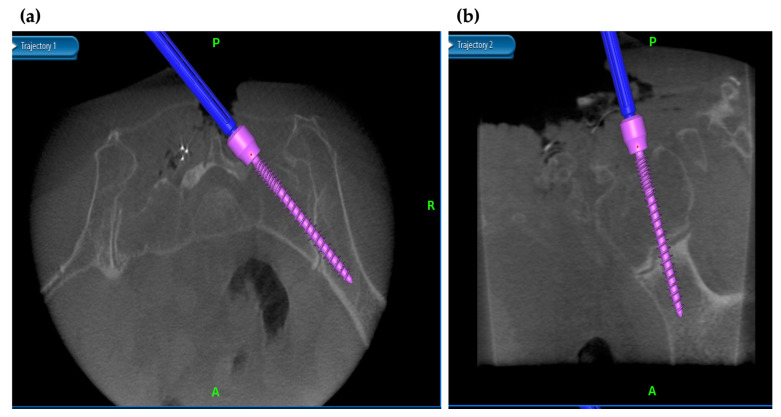
Navigation in minimally invasive spinal stabilization. With navigation, S2AI screws of sufficient length can be inserted in the exact direction. (**a**) Intraoperative axial and (**b**) intraoperative sagittal views on navigation monitor.

## Data Availability

Data sharing not applicable.

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
