# Peer review of "Computed Tomography-Based Navigation System in Current Spine Surgery: A Narrative Review"

_medicina, 2022, doi:10.3390/medicina58020241_

Round 1
Reviewer 1 Report
The authors chose an interesting topic recounted using a narrative review methodology. The article could be greatly improved if the authors abide by guidelines for narrative reviews. Starting with the title which should clearly state the methodology. "Computed tomography-based navigation system in spine surgery: A narrative review." For the other issues we recommend the following article - https://www.ncbi.nlm.nih.gov/pmc/articles/PMC2647067/
Many statements/claims require references:
Line 33
Lines 40-42
Lines 42-43
Lines 54-57
Lines 58-68
Lines 279-295 (There is an entire special issue in the Journal of Neurosurgery on Augmented and Virtual Reality in spine surgery which can help support these claims - https://thejns.org/focus/view/journals/neurosurg-focus/51/2/neurosurg-focus.51.issue-2.xml)
The manuscript has a few grammatical mistakes all through. These include the first sentence of the abstract "The number of spine surgeries using instrumentation has been increasing associated with recent surgical techniques and spinal implants." instead of "Instrumentation during spine surgeries has increased as a result of novel surgical techniques and spinal implants." Or in the trends section when the authors say "Navigation systems were firstly reported in the field of neurosurgery since 1970s" instead of "Navigation systems were first reported in the field of neurosurgery in the 1970s"
We recommend that the authors proofread their work with a software like Grammarly (https://app.grammarly.com/)
Author Response
The authors chose an interesting topic recounted using a narrative review methodology. The article could be greatly improved if the authors abide by guidelines for narrative reviews. Starting with the title which should clearly state the methodology. "Computed tomography-based navigation system in spine surgery: A narrative review." For the other issues we recommend the following article - https://www.ncbi.nlm.nih.gov/pmc/articles/PMC2647067/
Author’s response:
Thank you for your helpful comment. We revised the title as follows: “Computed tomography-based navigation system in spine surgery: A narrative review”. We also followed the method of narrative review you suggested and added a Methods and Discussion section. I have incorporated main part of the previous text into the discussion part.
Many statements/claims require references:
Line 33 (ref: DOI: 10.1097/01.BRS.0000076895.52418.5E)
Lines 40-42 (ref: DOI: 10.1016/j.injury.2004.05.009)
Lines 42-43 (ref: DOI: 10.3171/2012.5.SPINE11399)
Lines 54-57 (ref: DOI: 10.3171/2014.1.FOCUS13531)
Lines 58-68 (ref: DOI: 10.1155/2016/5716235)
Lines 279-295 (There is an entire special issue in the Journal of Neurosurgery on Augmented and Virtual Reality in spine surgery which can help support these claims https://thejns.org/focus/view/journals/neurosurg-focus/51/2/neurosurg-focus.51.issue-2.xml)
Author’s response:
Thank you for your comment. As you pointed out, we added the references as appropriate.
Line 38 (ref: DOI: 10.1097/01.BRS.0000076895.52418.5E)
Lines 43-46 (ref: DOI: 10.1016/j.injury.2004.05.009)
Lines 46-47 (ref: DOI: 10.3171/2012.5.SPINE11399)
Lines 63-66 (ref: DOI: 10.3171/2014.1.FOCUS13531)
Lines 67-77 (ref: DOI: 10.1155/2016/5716235)
Lines 295-312 (ref: DOI: 10.3171/2021.5.FOCUS21217, DOI: 10.3171/2021.5.FOCUS21209)
The last part pointed out has been partially modified based on the reference.
The manuscript has a few grammatical mistakes all through. These include the first sentence of the abstract "The number of spine surgeries using instrumentation has been increasing associated with recent surgical techniques and spinal implants." instead of "Instrumentation during spine surgeries has increased as a result of novel surgical techniques and spinal implants." Or in the trends section when the authors say "Navigation systems were firstly reported in the field of neurosurgery since 1970s" instead of "Navigation systems were first reported in the field of neurosurgery in the 1970s"
We recommend that the authors proofread their work with a software like Grammarly (https://app.grammarly.com/)
Author’s response:
Thank you for your kind comment. We revised our manuscript using Grammarly, and it was also edited entirely by a native English speaker.
Reviewer 2 Report
Well-presented manuscript, centered on the role of intra-operative navigation via the aid of computed tomography. A few remarks about your valuable effort are discussed below.
A language and Grammar check seems to be necessary, as a lot of errors are encountered. It would be valuable if you could insert data extracted from comparative studies, regarding the relevant efficacy and safety of computed tomography-guided instrumented fusion, in comparison with free hand techniques. There is paucity of data that could be extracted from large, single-center series or relevant review articles. This manuscript could more easily be concidered as an article that is under the term 'technical note' and not a 'review' type of manuscript.
Author Response
Well-presented manuscript centered on the role of intra-operative navigation via the aid of computed tomography. A few remarks about your valuable effort are discussed below.
A language and Grammar check seems to be necessary, as a lot of errors are encountered.
Author’s response:
Thank you for your helpful comment. The manuscript was edited entirely by a native English speaker.
It would be valuable if you could insert data extracted from comparative studies, regarding the relevant efficacy and safety of computed tomography-guided instrumented fusion, in comparison with freehand techniques. There is a paucity of data that could be extracted from large, single-center series or relevant review articles.
Author’s response:
We appreciate your comment. As you suggested, we searched the data regarding the relevant efficacy and safety of computed tomography-guided instrumented fusion and added these data in “Accuracy, complication rate, cost-effectiveness, and radiation exposure in Navigated Spine Surgery” in the discussion as below.
Shin et al. reported that 94% of pedicles screws were inserted accurately with navigational techniques, while 85% were inserted accurately with freehand techniques. Verma et al. also reported 93.3% of the pedicle screws were inserted accurately with navigational techniques, while 84.7% were inserted accurately with freehand techniques. (lines 98-102)
This manuscript could more easily be considered as an article that is under the term 'technical note' and not a 'review' type of manuscript.
Author’s response:
Thank you for your suggestion. Another reviewer also kindly suggested us to change the type of paper from “review” to “narrative review”. We think it seems reasonable because this is mainly based on previous papers. Therefore, we would like to revise the type of paper to “narrative review”. If Editor prefers to change the type of paper to “technical note”, we will gladly follow his decision.
Reviewer 3 Report
The authors in this paper perform a non-systematic review of the literature on the uses of navigation systems in spinal surgery by associating it with minimally invasive surgery.
The paper although well written does not represent a novelty in the literature, navigation systems coupled with the radiological image acquired in real time overcome the problems encountered on interventions of mobile spinal segments such as the cervical spine.
Moreover, minimally invasive techniques only in rare occasions need to use navigation systems because with the correct technique in radioscopy there are no problems of screw malpositioning.
I recommend reject.
Author Response
The authors in this paper perform a non-systematic review of the literature on the uses of navigation systems in spinal surgery by associating it with minimally invasive surgery.
The paper although well written does not represent a novelty in the literature, navigation systems coupled with the radiological image acquired in real time overcome the problems encountered on interventions of mobile spinal segments such as the cervical spine.
Moreover, minimally invasive techniques only in rare occasions need to use navigation systems because with the correct technique in radioscopy there are no problems of screw malpositioning.
I recommend reject.
Author’s response:
We appreciate your comment. We understand your claim. This review was written in the style of narrative review to describe the history and current status of navigated spine surgeries comprehensively. Since another reviewer also kindly suggested us, we have mentioned that this article is a narrative review in the title. That said, we really appreciate your suggestions, and this will be a future challenge.
Although cervical spine surgeries and MIS surgeries can be performed using fluoroscopy, it is often difficult to visualize the lower cervical and upper thoracic spine. It is also difficult to obtain exact A-P and lateral views especially in the cases of spinal deformities, congenital malformation, and revision surgeries. It is sometimes impossible to adjust the positions of the patient and fluoroscopy to obtain exact views in the scoliotic and rotated spine. As we mention in the manuscript, there are many advantages of navigation systems to overcome these problems. In MIS spine surgeries, surgeons can realize the anatomy and correct trajectories of spinal implants even in a limited field of view. And also, percutaneous pedicle screws can be inserted in the cases of the upper thoracic spine, osteoporosis, and obese patients, which are difficult to visualize under fluoroscopy. To perfume various spine surgeries safely and accurately, we consider that it is reasonable to use the CT-associated navigation systems in the current situation. We believe that the limitations of navigation systems should also be described in the manuscript as well as their advantages to avoid serious complications. We hope that we can overcome the problems by knowing the current status of navigation systems. This is why we describe future challenges and prospects in this manuscript.
Round 2
Reviewer 1 Report
Congratulations. I look forward to reading your work in print.
Reviewer 2 Report
I appreciate your reply to the comments attributed to the submitted manuscript. I concider that the addition of the term 'narrative review' to the title of your manuscript is acceptable and describes with accuracy your work.
Reviewer 3 Report
Accepted